# Chemical Treatments on Invasive Bivalve, *Corbicula fluminea*

**DOI:** 10.3390/ani14121789

**Published:** 2024-06-14

**Authors:** Katie D. Goulder, Wai Hing Wong

**Affiliations:** 1Division of Wetlands and Waterways, Massachusetts Department of Environmental Protection, 100 Cambridge Street, Suite 900, Boston, MA 02114, USA; goulder.k@northeastern.edu; 2Environmental Science and Policy, College of Science, Northeastern University, 360 Huntington Avenue, Boston, MA 02115, USA

**Keywords:** Asian clam, chemical treatment, invasive species, *Corbicula fluminea*, bivalve mollusc, molluscicide, management, toxicity

## Abstract

**Simple Summary:**

The Asian clam, *Corbicula fluminea*, is a highly invasive aquatic mollusc that has become one of the most ecologically and economically harmful invasive species in aquatic ecosystems in Europe, North America, and South America. Due to the impacts of climate change, Asian clams are shifting their range farther north and are inhabiting waterbodies that were once too cold to survive. Chemical treatment has been found to be the most economical and effective management strategy for controlling invasive mollusc populations and preventing their further spread. However, chemical treatments must maximize Asian clam mortality while minimizing the impacts on water quality and non-target species. Although there are numerous treatments on the market, they are not widely applied in the field. A comprehensive review of Asian clam molluscicides was performed to evaluate the efficacy of currently available chemical treatments and the toxicity ranges of emerging treatments. The results of this review will aid resource managers in invasive Asian clam control and management.

**Abstract:**

The Asian clam *Corbicula fluminea* is a native aquatic species in Eastern Asia and Africa but has become one of the ecologically and economically harmful invasive species in aquatic ecosystems in Europe, North America, and South America. Due to their natural characteristics as a hermaphroditic species with a high fecundity and dispersal capacity, Asian clams are extremely difficult to eradicate once they have infiltrated a waterbody. This is an emerging issue for states in the Northeastern United States, as Asian clams expand their range farther North due to climate change. There has been extensive research conducted to develop chemical treatments for reactively controlling invasive mollusc populations and proactively preventing their further spread. However, treatments are mostly targeted toward biofouling bivalves in industrial settings. A comprehensive review of Asian clam chemical treatments used in natural open-water systems was performed to evaluate molluscicides and identify the toxicity ranges of emerging treatments that maximize Asian clam mortality and minimize the negative impact on water quality and non-target species. The potential chemical applications in Asian clam control and management are summarized in this report to assist resource managers and practitioners in invasive Asian clam management.

## 1. Introduction

### 1.1. Life History

The Asian clam is a small bivalve mollusc with a shell length less than 50 mm. Its shell has concentric sulcations, with serrated anterior and posterior lateral teeth [1]. The shell coloration appears to be latitude-dependent, with lighter morphs found in the northeast United States and darker morphs found in the southwestern United States. Asian clams have high physiological plasticity but prefer areas of slower water flow, and environmental factors such as temperature, food availability, and dissolved oxygen can affect individual growth rates and reproductive capacity [2,3,4]. The Asian clam is also a self-fertilizing hermaphroditic species that can produce up to 570 larvae per day, resulting in more than 68,000 larvae produced per an individual per year [5]. This, combined with their rapid growth and early sexual maturity, enables a single individual to initiate rapid reproduction in isolation [6,7]. In addition to their high fecundity, their extensive dispersal capacities due to their association with human activities makes the Asian clam a successful and highly invasive species that is difficult to control and manage [8,9,10,11,12].

Although native to Eastern Asia and Africa, Asian clams have invaded aquatic ecosystems in Europe, North America, and South America. Asian clams are thought to have originally been introduced to the West Coast of North America in 1938 through transoceanic ballast water exchange and Chinese immigrant importation for use as a food source [10,13,14]. By the 1960s, the Asian clam had spread throughout the United States’ waterways to reach the Atlantic coast [8]. Since then, Asian clams have established themselves above the 40° latitude line, previously thought to be the threshold of their low temperature tolerance [15].

### 1.2. Asian Clams Dispersal in Massachusetts, USA

Previous studies reported that the Asian clam was first sighted in Massachusetts in the Charles River in 2001. This is fitting, as the state’s likely origin of infestation is the Charles River, which is a central figure in the Eastern Massachusetts landscape and is a hotspot for boating year-round [15]. However, newly discovered historical data reveal that Asian clams were first detected in Massachusetts farther South in Long Pond at Lakeville as early as 1999 [16]. Long Pond is frequently used for recreational boating, fishing, and swimming, all of which are common vehicles for aquatic invasive species introduction and contamination. Before 2017, Asian clams were found in 36 waterbodies, mostly in the south and southeast of the state, with an abundance of up to 6124 clams/m^2^ recorded in an infested unnamed tributary in Forest Park, Springfield, Massachusetts [17]. Since their introduction in 1999, Asian clams have been found in 45 waterbodies, including 8 rivers and 37 ponds and lakes in Massachusetts [15]. In the past 5 years, Asian clams have expanded into waterbodies farther northwest within Massachusetts (Figure 1), where minimum water temperatures are usually lower. This northern expansion of range is consistent with other research, as the species’ presence has recently been detected in other New England states, such as Lake Bomoseen in the State of Vermont [15,18] and the lower Merrimack River and several ponds in the State of New Hampshire [15,19,20].

### 1.3. Impacts of Invasion

As climate change causes temperatures to continue to increase, Asian clams are likely to continue to expand their range farther north and represent an emerging issue for the Northeast United States. Not only are Asian clams the most widely distributed aquatic nuisance species in the contiguous United States, but they are also one of the most ecologically and economically impactful aquatic invasive species globally [8,10]. As suspension feeders with high filtration rates, Asian clams can drastically change the ecosystems through selective feeding, consuming large quantities of microscopic plants and animal species, greatly altering the macroinvertebrate communities [21,22]. This also leads to them outcompeting native unionid species for benthic microbes and causing up to a 70% reduction in phytoplankton [22,23]. In addition to direct competition for space and food, Asian clams can also impair native freshwater mussel reproduction and recruitment by filtering sperm from the water column or ingesting unionid glochidia directly. They also displace native unionid mussels through abiotic changes such as the sequestration of large amounts of available carbon in the water and the production of high levels of toxic ammonia through feces excretion and mass die-off events [22,23]. Finally, Asian clams’ mode of life of burrowing in the sand instead of attaching to hard substrates can lead to bioturbation that may negatively impact native unionid filtration [21]. As unionid mussels are among the most endangered faunal groups, Asian clam invasions are of particular concern [24,25,26].

In addition to their ecological impacts, Asian clams also have a significant economic impact due to the loss of ecosystem services and direct economic damages. As Asian clams are introduced to new waterways by human activity, they can damage or occlude equipment such as water intake pipes in electrical power plant cooling systems and sewage treatment plants [15,24,26]. Although the real cost is not known, the costs needed to control Asian clam populations across the United States have been estimated to be over USD 1 billion annually since 1980 [27]. As Asian clams continue to expand their range and infest new waterways, their ecological and economic impact is expected to increase [15].

### 1.4. Management Strategies

While the prevention of aquatic invasive species infestations is the ideal management strategy, an integrated approach is necessary to best treat waterways once Asian clams have been detected to prevent further spread. Wong [15] notes that this integrated strategy should contain four steps:Identify the primary means of spread.The most vulnerable waterbodies must also be identified to allocate resources in the most cost-effective manner.Create a plan to stop or slow the spread.Public awareness plays a large role in limiting dispersal, so developing educational content and signage should be a priority. There must also be an investment in infrastructure to prevent spread, such as installing washing stations at susceptible waterbodies.Analyze the ecological and economic impact of infestations on new water bodies.Create an emergency response plan to proactively limit the species’ invasiveness.

The emergency response plan will contain components of all four steps outlined above, but it will focus primarily on the active treatment. These methods can be both reactive, once a population of Asian clams has been detected, and proactive, targeting Asian clams in the larval stage before they have been established. Proactive treatments tend to be more effective, versatile, and cost-effective than reactive measures. However, due to limited resources, they are often not employed until after infestations have been detected [15,28]. Treatment options include physical, biological, genetic, and chemical options, as depicted in Table 1. While each treatment type can be employed individually, there is potential to combine techniques to enhance their efficacy.

### 1.5. Treatment Options

#### 1.5.1. Physical Treatments

Physical treatments alter the Asian clam’s habitat to make it less suitable for their survival and include altering the temperature or dissolved oxygen content of the water. Gas-impermeable benthic barriers have been proven effective at depleting dissolved oxygen and culling Asian clam populations. However, they are only a short-term control strategy and are often detrimental to non-target benthic species [15,28,29]. Thermal shock, either through dry ice or open flame, has also been effective at treating Asian clams but is not always feasible under natural conditions. Physical removal, either through suction dredging or hand removal, remains the most effective form of physical control but is costly and only targets adult life stages [15].

#### 1.5.2. Biological Treatments

Biological treatment options are more limited because although Asian clams are small in size, their shells are proportionately quite strong, and few predators are able to consume them. Parasites or disease vectors may be more effective at controlling Asian clam populations, but few options are currently available on the market [15]. As a biological treatment, Zequanox^®^ induces mortality by damaging the epithelial cells in the intestinal lining following ingestion [37].

#### 1.5.3. Genetic Treatments

Newer technology has been introduced to prevent invasive species from reproducing by either manipulating genes to reduce fecundity or by using X-rays to prevent the development of the zygote. While it may be a promising tool for other bivalves, for which this could be used both as proactive and reactive management, it is expensive, and laboratory testing was not able to achieve 100% success in preventing further spread [36]. Additionally, it may not work on the Asian clam due to its unique reproduction strategy. Further research is needed to determine whether or not radiation such as X-rays can work on treating Asian clams and how to increase efficacy and perform this technique at scale in a cost-effective manner.

#### 1.5.4. Chemical Treatments

The use of chemicals to treat a waterbody remains one of the only forms of proactive treatment and the most efficacious and cost-effective method for Asian clam eradication, suppression, and control. However, it also has the greatest potential to negatively impact non-target species and the surrounding environment [15].

This analysis will focus on studies on Asian clam chemical treatments currently available for applications. It will also evaluate treatments and toxicities that have been tested only in a laboratory setting or on other invasive bivalves and have the potential for success in Asian clam control.

### 1.6. Treatment Selection

Chemical treatments have the largest range of options available that allow for site-specific Asian clam control plans. A variety of factors must be considered when determining which chemical treatment to use to achieve the goals of the project. A summary of the most crucial factors to consider when choosing a chemical treatment are listed below in Table 2. However, special attention is needed to determine whether a water system is considered open or closed. Closed systems include water intake pipes, wastewater treatment facilities, and electric powerplants cooling systems, where the used water is not discharged back into the waterbody source [17]. Treatment is aimed at eradicating existing biofouling clams or preventing biofouling clams from accumulating on these commercial structures. Due to their great potential economic impact, many treatments have been specifically designed to be used in these industrial settings [28]. Non-target species are of minimal concern in this setting, so more aggressive control strategies can be employed. In these closed systems, usually, oxidative chemicals such as chlorine or bromide are used to eradicate the invasive bivalves quickly and effectively. However, chlorine and bromide produce carcinogenic byproducts and halogens that are toxic to non-target species [32,37]. To prevent environmental contamination, chemical treatments used to prevent or treat biofouling clams in closed systems are not suitable chemical treatments for open-water systems and natural water bodies [29].

It is more difficult to stop the spread of Asian clams in open-water systems, where treatment methods are more restricted due to their potential to negatively impact water quality, including drinking water contamination, and non-target species [15]. Additional consideration and extensive testing protocols must be employed to ensure that the chemical treatment used to eradicate invasive Asian clams does not have deleterious environmental consequences. For the purposes of this review, only treatments that have been used or have the potential to be used in natural open-water systems have been included in the analysis. Some of the most common active ingredients used in molluscicides for Asian clam chemical treatment are copper-based compounds. The copper-based treatments induce mortality in Asian clams through binding to receptors on the bivalves’ gills, thus inhibiting gas exchange across the cellular membrane [15,38].

### 1.7. Regulations

The chemical treatment selected must be approved by the regulatory body in the country or state of use. The European Union Regulation number 1143/2014 adheres to hierarchical measures of prevention, early detection and rapid eradication, and management to combat invasive species [39]. In the United States, the Environmental Protection Agency (EPA) sets the standard for molluscicide use either by regulating specific products or granting special permits for emergency circumstances. This review will focus on chemical treatments used in the United States, but specific consideration for more local regulations on approved molluscicides and the regulations regarding the potential water quality and environmental impacts must be strictly adhered to.

The Federal Insecticide, Fungicide, and Rodenticide Act (FIFRA) is the primary EPA regulation for chemical treatments used as molluscicides. Currently, brand-named Natrix^®^ (Carmel, IN, USA) and EarthTec QZ^®^ (Rogers, AK, USA) are copper-based products registered with the EPA as molluscicides, although Cutrine^®^ Ultra (Carmel, IN, USA) has been registered with the EPA as an algaecide, herbicide, and cyanobactericide. KCl, or unrefined potash, is not registered with the EPA as a molluscicide but can instead be approved for use through site-specific regulatory exemption processes [38]. Section 24© of FIFRA allows states to register a pesticide for a special local need, defined as “an existing or imminent pest problem for which there is no federally registered pesticide available” [38,41]. In addition, FIFRA Section 18 “allows unregistered use of pesticides to address emergency conditions” [38,42]. Examples of use in projects for rapid response zebra mussel eradication include Lake Irene and Rose Lake in Minnesota in 2011, and examples for use in population suppression include Lake Ossawinamakee in Minnesota in 2004 and 2005 [38].

In addition to specific molluscicide regulations, regulations involving the discharge of chemicals into water bodies must also be adhered to. The Clean Water Act regulates all molluscicides and biocides discharged into US waters. Chemical treatments must be registered with the United States EPA and handled and applied according to label instructions [32]. Chemical treatments may also be subject to different regulations in different countries or states/provinces. For example, if chemicals are applied to bodies of water in Massachusetts, USA, for the control of aquatic nuisance species, a WM04 Chemical Application License needs to be granted by the Massachusetts Department of Environmental Protection, per Massachusetts General Laws c. 111, s. 5E. More detailed information can be found in the book titled “Invasive Animals and Plants in Massachusetts Lakes and Rivers: Lessons for International Aquatic Management” [15].

## 2. Methods

### Systematic Review

A variety of Asian clam control methods have been developed for practitioners, including physical, biological, and chemical control. This review will focus primarily on chemical control, as this strategy requires natural resource managers to use the fewest resources while achieving the greatest impact. A review was conducted using a variety of sources to obtain the most comprehensive collection of data currently available on the chemical treatment of Asian clams and similarly invasive bivalves with applications in natural, open-water systems. Sources included Google Scholar, using the keywords “Asian clam Chemical Treatment” to identify peer-reviewed articles, state and federal regulatory documents, and chemical product safety data sheets through March 2024. In addition to these documents, the Invasive Mussel Collaborative website was used to identify primary, peer-reviewed sources. The sources were then categorized by subject matter, determined by whether reported studies were conducted on Asian clams in the field, were Asian clam toxicity studies in a laboratory setting, or focused on the treatment of other species of invasive bivalves, which treatment could potentially be applied to Asian clams in future studies (Figure 2). The following results are categorized as such.

## 3. Results

### 3.1. Applied Chemical Treatments

Despite their market availability, chemical treatments have not been widely applied to Asian clam infestations in natural waterbodies. This is most likely due to the lack of public and resources manager education and awareness regarding the impacts of this invasive species. In addition, as Asian clams inhabit the substrate at the bottom of a waterbody, they usually do not impact popular recreational activities, such as swimming and boating. Invasive plants, such as Eurasian milfoil or curly-leaf pondweed, are more exposed and therefore garner more public attention and responses from resource managers, as these species can impair the general public’s recreation. Asian clams that infiltrate industrial equipment or intakes, known as biofouling, cause the greatest economic impact, and as such, chemical treatments have focused on Asian clam eradication in this setting [4]. Oxidative treatments, such as chlorination and bromination, are the most common treatments used to treat Asian clams. Chlorine treatments have had success inducing the mortality of biofouling Asian clams in industrial, closed-water systems, such as hydroelectric dams and water treatment facilities. However, oxidizing treatments are not suitable for use in natural, open-water systems due to their lethal effects on non-target species in low concentrations [30]. As such, they will be omitted from this review.

### 3.2. Toxicity Studies

Apart from existing chemical applications, there are numerous toxicity studies on Asian clams, and some of these treatments may have the potential to be applied as new methods in the future. These molluscicides and their concentrations under various dissolved oxygen concentrations are listed in Table 3, and the original studies are described below.

#### 3.2.1. Asian Clam Sensitivity in Laboratory Testing

Wild-caught adult Asian clams were acclimated to municipal water maintained at 20 ± 2 °C with a 16 hr^L^ and 8 hr^D^ photoperiod cycle with continuous aeration. Mortality tests were then conducted under a geometric range of concentrations for each of the six chemicals. Each treatment was conducted under both normoxic conditions (>7 mg/L dissolved oxygen) and hypoxic conditions where nitrogen gas was infused into the water to decrease the dissolved oxygen content to less than 2 mg/L. While Asian clams display great physiological plasticity and are able to acclimate to various environmental conditions, they are largely intolerant to low oxygen levels. The results of this study demonstrate that the hypoxic conditions increased the susceptibility of clams to the chemical treatment by up to 400% relative to the same treatments performed under normoxic conditions [4]. The concentrations for the minimum effective concentration as well as the most effective concentrations of each chemical treatment under both hypoxic and normoxic conditions are summarized in Table 3.

The results of this study have important implications for managing invasive Asian clam populations. This toxicity study provides target concentrations for a variety of chemical treatments options. However, it also highlights how natural resource managers can combine treatment options to optimize results. Combining the physical and chemical management of invasive species could increase the effects of chemical treatment while decreasing the concentration of chemicals used, thereby minimizing the deleterious effects on non-target species and the surrounding environment. The use of curtain barriers reduces the dissolved oxygen content, increases Asian clam oxidative stress, and boosts the efficacy of chemical treatments. Similarly, the use of benthic barriers over treatment areas could help to induce hypoxic conditions to increase the treatment efficacy and clam mortality [31].

#### 3.2.2. Potassium Chloride and Formalin

In a study by Layhee et al. [48], a laboratory toxicity study was performed to determine if the molluscicide of 750 mg/L KCl and 25 mg/L Formalin (37% formaldehyde) used to treat zebra mussel veligers in hatcheries had similar success in Asian clams. Using this process, commonly referred to as Edward’s protocol [39], the study determined that there was a 100% mortality of Asian clam veligers after 3 and 5 h exposure times [48]. While further testing is needed to determine how this treatment affects different life stages as well as its potential for use in open-water systems, this initial study is promising for the prospect that chemical treatments used in the management of other invasive bivalves can successfully be used to treat Asian clams.

### 3.3. Chemical Treatments on Other Bivalves

Numerous molluscicides have already been developed to induce mortality in other invasive bivalves, such as zebra, quagga, and golden mussels. There is potential to expand the targeted species to include Asian clams following laboratory toxicity and field testing. Table 4 is a representative list of chemical treatments that are currently used to treat other invasive bivalves and have potential to be used in future Asian clam chemical treatment studies.

#### 3.3.1. Zequanox^®^

Another potential treatment for Asian clams is a biopesticide under the brand name Zequanox^®^ (Davis, CA, USA), which contains *Pseudomonas fluorescens* strain CL145A cells in a spent fermentation media. This bacterium naturally occurs in soil and water and is ingested by filter feeders, subsequently destroying their digestive lining and selectively killing dreissenid mussels [32,49]. While current studies discuss the effectiveness of Zequanox^®^ in controlling zebra and quagga mussels, evidence suggests it could be applied to treat Asian clams as well. Bolam et al. [40] investigated the feeding rates and prey selection of Asian clams and determined that they prefer diatoms in warmer months but switch their diet to preferentially feed on flagellates in the fall [40]. The bacterium *Pseudomonas fluorescens* in Zequanox^®^ is a flagellate bacterium, and thus, this product has the potential to be an effective treatment for invasive Asian clams, specifically during the fall months. Further testing is required to determine if Asian clams will preferentially filter out the Zequanox^®^ bacterium to ingest, what life stages it will treat, and what concentrations are required for the population density and feeding rates of a detected population during different seasons.

#### 3.3.2. Combination of Zequanox^®^, EarthTec QZ^®^, and Potash (KCl)

Treatments can be combined to facilitate a rapid response to the early detection of invasive bivalves. Such was the case in Christmas Lake, Minnesota when, in 2014, a localized population of zebra mussels was detected near the public boat landing. A combination of Zequanox^®^, EarthTec QZ^®^, and potash was used in successional treatments, with Zequanox^®^ used first in September 2014, only 23 days after the initial infestation detection. This instance also represents the first time Zequanox^®^ that had been employed as a treatment for Zebra mussels in the field. In October and November of the same year, a series of EarthTec QZ^®^ treatments were used in the initial detection area and in the area immediately surrounding it, although no curtain barrier was installed until the second treatment. Finally, in December 2014 and in April and May of the following year, potash was used to maintain a concentration of 73.5 and 110 ppm K^+^ within the treatment area for 10 days. Although these treatments resulted in no detected surviving Zebra mussels, individuals were found a year later in untreated parts of the lake. While this study represents an immediate response with early success, it also highlights the need for post-treatment monitoring and the limitations of treatments in only a portion of a water body [31]. The use of chemical treatment combinations and the lessons learned from this field study should be applied to treatments of all invasive bivalves, including Asian clams, as field studies continue to increase in frequency.

#### 3.3.3. Cutrine^®^ Ultra

Chemical treatments can be targeted towards different life stages by varying the concentration of the molluscicide used. A study by Kennedy et al. [53] sought to identify the most sensitive life stage of zebra mussels to optimize the chemical’s toxicity while minimizing the chemicals released into the environment and, subsequently, the impacts on non-target species. Toxicity tests of the copper-based algaecide Cutrine^®^ Ultra on early-life-stage and adult zebra mussels determined that early life stages were much more sensitive, and the 72-hr trochophores had an LC_99_ value of only 0.012 mg/L after less than 2 h. Adult zebra mussels were more resistant, but the 9-h LC_99_ value was 1.757 mg/L, and higher concentrations of Cutrine^®^ Ultra were required for shorter exposure times [53]. Further research is needed to determine the toxicity of chemical treatments for invasive bivalves at different life stages to increase their efficacy and achieve the more targeted control of invasive species populations.

#### 3.3.4. Microencapsulation

Microencapsulation represents a promising technology that could be applied to a variety of active ingredients already approved and used in molluscicides to treat invasive bivalves [37]. Microencapsulation technology encases the control agent in an edible coating to bypass a bivalve’s shell-closing response. Instead of sensing a harsh chemical such as chlorine and closing its shell for 2–3 weeks at a time, the encapsulated active ingredient is ingested quickly during filter feeding and, thus, is taken up more rapidly than harsh chemicals [34]. As the active ingredient is selectively absorbed by the target species, microencapsulation requires lower concentrations of active ingredients while increasing its toxicity. This allows for more effective treatment while also reducing the residual environmental contamination [37].

BioBullets^®^ (London, England) are a manufactured lipid-walled microparticle that encases various control agents. In a report by Tang and Aldridge [56], two new formulations of BioBullets^®^ with the smallest particle size most similar to the filter feeder’s prey were tested on *Rangia cuneata* or Wedge clams. The SB 1000 model had a particle size of 9.5 ± 0.5 µm with a smooth surface, and the SB 2000 model had a particle size of 24.6 ± 1.3 µm with a crystallized surface. Asian clams have a prey particle size preference of <20–25 µm, so both models are likely to be ingested by Asian clams. Despite their small size, SB 2000 had an extended-release rate over 20 h. A single dose of 2–6 mg/L of the active ingredients resulted in 90% mortality within 30 days [56]. While the study does not specify what the active ingredients are in the tested BioBullets^®^, it does refer to SB 1000 as having a cationic polymer that acts as a surfactant, which is most likely poly-diallyldimethyl ammonium chloride (polyDADMAC), used in previously reported studies [4], while the SB 2000 BioBullets^®^ used an anionic salt, most likely KCl. While both active ingredients are toxic to bivalves, Calazans et al. [37] reported that encapsulated polyDADMAC was 9.3 times more toxic than encapsulated KCl in golden mussels [37]. As the prey particle size is within the preferred range and the various models with differing surface textures could be used according to their seasonal diet preferences, BioBullets^®^ have great potential for use as Asian clam chemical treatments [56].

#### 3.3.5. Essential Oils

There is rising interest in essential oils and their constituents as a safer alternative to pesticides used to treat aquatic gastropods. To determine the immersion lethal toxicity value of various essential oils, researchers immersed adults and egg clusters of freshwater *Pomacea canaliculata* snails in essential oils for 24 h, followed by a recovery period in dechlorinated tap water. After a 12 h exposure period, 100% mortality was achieved at 156.25 µg/mL. However, this review also showed that essential oil constituents were more deadly to aquatic gastropods when ingested through baiting as opposed to diffuse essential oils released into the water column. Specifically, feeding on baits that contained sublethal concentrations of eugenol, a compound found in cinnamon, clove, and bay leaves, the 24-h lethal concentration of 50% of the sample size (24 h LC50) ranged between 2.55 and 10.73 µg/mL, with the highest toxicity occurring in conjunction with elevated temperatures. Eugenol also reduced the fecundity, hatchability, and survival of young *Lymnaea acuminata* [35]. Although Asian clams are different from gastropods, there is potential to combine essential oil baiting techniques with microencapsulation technology to enable invasive bivalves to filter out and consume the toxic essential oil constituents and maximize their efficacy.

Although the mechanism of toxic action remains unknown, it has been proposed that the various constituents of essential oils interfere with metabolic, physiological, and behavioral functions [35]. Further research is needed to identify the mode of action to determine the most effective formulation and delivery system for reaching the target species. Site-specific toxicity testing is also needed to determine the impacts of essential oils on non-target species.

## 4. Future Directions

To date, there are no known control methods that have achieved the complete eradication of Asian clams in natural open waterbodies. Only opportunistic studies have occurred as invasive bivalves are detected, and paid responses have been deployed [31]. However, infestations in industrial settings have been extirpated, and there is increasing research on invasive bivalve population responses to molluscicides that have promising results [30].

The tables presented in this review outline the potential for future studies. Toxicity studies (Table 3) should be tested in the field to determine if they could be candidates for permittable Asian clam treatments. The compounds in Table 4 represent chemical treatments that have proven successful in treating other invasive bivalves such as zebra and quagga mussels and have the potential to effectively treat Asian clams as well. Further testing is needed to first determine the toxicity concentrations that induce mortality in Asian clams in a laboratory setting, followed by field testing to determine the environmental impacts of releasing molluscicides into infested waterbodies. Pre- and post-treatment monitoring is required to determine the impacts of chemical treatments on all life stages of Asian clams, non-target species, and water quality data as well as efficacy-related data that will help to inform safe and effective adaptive management [38].

Finally, future research should investigate how best to combine chemical treatments with other physical or biological treatments to maximize efficacy. As shown in the study by Rosa et al. [4], inducing hypoxic conditions with the use of benthic barriers increased the Asian clam mortality, using lower concentrations of each chemical treatment [4]. Combining treatment methods to manage invasive species could help to increase the effects of a chemical treatment while decreasing the concentration of chemicals used, thereby decreasing the potential impacts on the surrounding environment and non-target species.

## 5. Conclusions

Asian clams’ high fecundity and physiologic plasticity have allowed them to become a highly successful invasive species in the North America. As Asian clams expand their range to infest waterbodies extending farther into the Northeast United States, effective control strategies are needed to help manage these invasive populations to prevent ecological and economic harm. While physical, biological, and genetic control strategies present opportunities for effective invasive bivalve control, chemical treatments are one of the only preventative control strategies currently available at scale. They are also the most cost-effective management option and effectively treat all life stages of this species, although careful consideration of variables such as the habitat and water quality of the proposed treatment area is needed to determine the most suitable molluscicide. Despite multiple molluscicides being available on the market, their use to eradicate Asian clam infestations in natural waterbodies is quite limited, most likely due to the lack of awareness regarding their ecological and economic impacts. Chemical treatments have undergone lab testing to determine the minimum concentrations needed to initiate Asian clam mortality, as well as the concentrations that cause 100% mortality under hypoxic and normoxic conditions [4]. Finally, chemical treatments used to eradicate other invasive bivalves, including zebra and quagga mussels, have been identified. Both the laboratory toxicity testing and the treatments of other invasive bivalves require further research to determine their efficacy in treating Asian clams both individually and in tandem with other forms of management. This review summarizes the current available research on the chemical treatment of Asian clams to direct future research efforts and to guide resource managers and practitioners on invasive Asian clam management.

## Figures and Tables

**Figure 1 animals-14-01789-f001:**
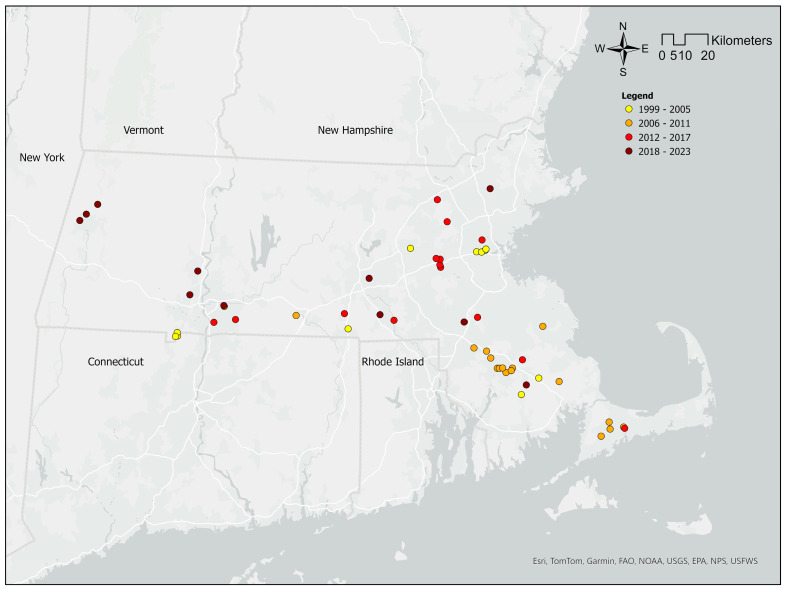
Distribution map depicting *Corbicula fluminea* in Massachusetts. Sightings were grouped by first reported sight years and color-coded to demonstrate the northwestern expansion of their range over 24 years (see Appendix A for site descriptions).

**Figure 2 animals-14-01789-f002:**
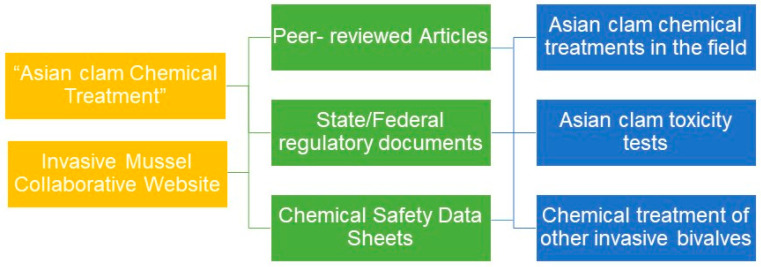
Workflow of the systematic review categorization process.

**Table 1 animals-14-01789-t001:** Evaluation characteristics of each option available for treating Asian clams.

Types of Control	Management Technique	Strategy Options	Pros	Cons	Source
Physical	Reactive	Gas Impermeable Benthic Barrier	Minimal impact to free-swimming, non-target species	Resource intensive: Time and money	[15,28,29]
			Can achieve 100% mortality	Bottom substrate could impede barrier	
				Negatively impacts non-target bivalves	
				Short term control strategy	
				Treats only adult life stage	
		Hand Harvesting	Minimal impact to non-target species	Resource intensive: Time and money	[15]
				Treats only adult life stage	
		Suction Dredging	Capable of removing larger quantities of clams at once	Disruptive of aquatic habitat	[15,30]
				Negative impacts to non-target species	
				Treats only adult life stage	
		Water Level Control; Drawdown	Treats all life stages of clams	Only feasible in small ponds; Not feasible in large waterbodies or rivers	[15]
				More effective in colder climates	
		Thermal Control; Dry Ice, Open Flame, Steam	Commercially available dry ice pellets	Laboratory results only	[12]
			Can be used directly (water absent) or indirectly (clams submerged)	Did not achieve 100% mortality	
			Treats all life stages of clams	Open flame only available for emersed substrate	
Biological	Reactive	Bacteria-based Molluscicide	Highly specific to target species	Target species currently limited to zebra mussels and quagga mussels	[17,28,31,32]
				Unknown reactions with non-target species	
				Use in open water systems is debatable	
		Predators (ex. barbel and crayfish)	Minimal impact to non-target species	Cannot eliminate or sustainably control invasive bivalves alone	[15,30,33]
				Harder shells make them less palatable to most predators	
				Treats only adult life stage	
Chemical	Proactive and Reactive	Single Synthetic Chemical	Most commonly used treatment	Negative impacts to non-target species	[15,30]
			Treats all life stages of clams		
			Most cost effective		
		Encapsulated Formulations	More targeted treatment requires less product use	Treats only adult life stage	[15,17,28,34]
				Emerging technology; few products available	
		Plant Extracts	Chemical complexity make it difficult to become resistant	Mode of application more effective for terrestrial or mobile aquatic gastropods	[35]
			Versatile uses	Mechanism of toxic action unknown	
			Rapid degradation prevents persistence and bioaccumulation	Easily oxidized by heat, light, and air	
			Non-toxic to non-target species and environment	Could be cost prohibitive in large volumes needed	
				More effective treatment for adult life stages	
Genetic	Proactive	Gene Manipulation	Proactive and preventative measure	Not currently a technique available for Asian clams	[36]
			Reduces mollusc fecundity	Resource intensive: Time and money	
				Results are inconclusive	
				Only feasible to treat adult life stage	
	Reactive	X-ray irradiation	Decrease development between zygote and trochophore stage	Limited induced mortality	
				Not able to fully sterilize males (Quagga mussels)	
				Only feasible to treat adult life stage	

**Table 2 animals-14-01789-t002:** Variables to consider when selecting a control strategy for Asian clams.

Type of Variable	Variables	Impact on Treatment	Source
Regulatory Variables	Chemicals permitted for use as molluscicide	Can vary by country, state, treatment area	[32,38]
Climate Variables	Minimum Temperature During the Coldest Months	Minimum temperature threshold and range shifts due to climate change	[17]
Habitat Variables	Type of Waterbody (open/closed)	Permitting of chemical treatment varies	[32]
	Treatment area size	Some treatments can be cost prohibitive depending on volume of chemical used	[31]
	Location along tributary	Prefer upstream habitats but must also consider downstream effects of treatment	[17]
	Water flow rate (River)	How to maintain effective concentrations throughout treatment area	[17]
	Water Mixing (Lake/pond)	How to distribute product to maintain effective concentrations throughout treatment area	[31]
	Bottom Substrate	Prefer coarse sand and soft mud	[17]
	Bathymetry	Treatment denser than water may result in “hotspots” at greatest depths	[31]
	Non-target Species Presence	Toxicity of non-target species must be considered to avoid causing unnecessary harm	[15]
Water Quality Variables	Temperature	May impact the efficacy of the treatment (Zequanox, KCl, Copper-based pesticides)	[38,39]
	pH	May impact the efficacy of the treatment (KCl and Copper-based pesticides)	[17,32,38]
	Salinity	May impact the efficacy of the treatment (KCl)	[38]
	Turbidity	May impact the efficacy of the treatment (Zequanox, KCl, Copper-based pesticides)	[38]
	Dissolved Oxygen Content	May impact the efficacy of the treatment (KCl, Copper-based pesticides)	[4]
	Conductance	May impact the efficacy of the treatment (KCl)	[32,38,39]
	Organic Carbon Content	May impact the efficacy of the treatment (Copper-based pasticides)	[32,38]
Clam Behavioral Variables	Time of Year	Feed less in colder months- may impact efficacy of ingested molluscicides	[31,40]
	Food Preference	Changes seasonally- may impact efficacy of ingested molluscicides	[40]
Project Goals	Reactive or Proactive	Resources required may vary	[32]
	Management, Suppression, or Extirpation	Determine how aggressive treatments should be	[31]

**Table 3 animals-14-01789-t003:** Toxicity of common active ingredients in molluscicides for Asian clams (NA = Not Applicable).

Chemical	Brand Name	Effective Minimum Concentration (mg/L)	In Vivo Concentration under *hypoxic* Conditions < 2 mg/L Dissolved Oxygen) (mg/L)	In vivo Concentration under *normoxic* Conditions (>7 mg/L Dissolved Oxygen) (mg/L)	Exposure Time	LC_50_ Concentration (mg/L) under Normoxic Conditions [95% CI in Brackets]	Modes of Action	Advantages	Disadvantages	Notes	Source
Ammonium Nitrate		59.3	50–200	125–200	96 h	201.00 [185.95–218.20]	Damages the gill epithelium to cause asphyxiation	Already used in agricultural channels as it is commonly used in agricultural fertilizer	Non-target gammarid and amphibian species are very sensitive to exposure		[4,15,28,43]
							Promotes glycolysis and oxidative phosphorylation	Economical			
								Non-toxic to most non-target species			
Chlorpyrifos		0.05	NA	0.5–1.0	96 h	No mortality was observed	Reduces cholinesterase activity	Agricultural chemical already in use	Lethal and sub-lethal effects of chemical on freshwater bivalves is largely unknown		[4,15,44]
							Reduces ability to burrow into the substrate	Exposure to chemical likely already occurring due to Asian clam proximity to agricultural fields			
Dimethoate		18	45–100	150–400	96 h	367.70 [325.68–429.79]	Inhibits acetylcholinesterase activity causing death or sub-lethal neurophysical effects including reduced burrowing capacity and alters bivalve closure behavior	Quickly degraded and non persistent in environment	Lethal to non-target species at low concentrations	Never reached 100% mortality	[4,15,45]
Niclosamide	Bayluscide	0.08	0.1–0.5	0.2–1.2	96 h	0.46 [0.38–0.55]	Induces mitochondrial fragmentation	Requires lowest concentration of biocide to produce lethal effects	Non-selective and negatively impacts non-target species		[4,15,28]
							Contributes to apoptotc and autophagic cell death	Short-lived in water	Use in open water is highly restricted		
polyDADMAC		12.3	10–500	10–1200	96 h	108.68 [76.44–153.94]	Disrupts membrane transfer mechanisms including gas exchange in the gills	Multiple functions chemical	Lethal to non-target species at low concentrations		[4,15,46]
			Peak at 200	Peak at 200				More suitable for use in closed systems			
Potassium Chloride (KCl)	Potash	45	100–250	150–900	96 h	NA	Induces paralysis and alters behavior of closure behavior	Minimal impact to non-target species	Not for use in waterbodies with high algae and macrophyte concentrations		[4,15,47,48]
				Peak at 450			Promotes ciliostasis to reduce bivalve filtration rate		Volume needed to maintain efficacious concentrations could be cost prohibitive in large area		
Potassium Chloride (KCl) and Formalin		NA	NA	750 KCl & 25 formalin	3–5 h	NA	Induces paralysis and alters behavior of closure behavior	Minimal impact to non-target species	Not for use in waterbodies with high algae and macrophyte concentrations	Only tested on Asian clam veligers life stage	48
							Promotes ciliostasis to reduce bivalve filtration rate	Treatment commonly used to kill zebra mussel veligers in hatchery trucks	Volume needed to maintain efficacious concentrations could be cost prohibitive in large area		

**Table 4 animals-14-01789-t004:** Chemical Treatments of other Invasive Bivalves with potential applications for Asian clam management. (* represents values estimated based on figures available in respective studies. NA = Not Applicable).

Chemical	Brand Name	Species Tested	Effective Minimum Concentration (mg/L)	In Vivo Concentration under *normoxic* Conditions (>7 mg/L Dissolved Oxygen) (mg/L)	Exposure Time	Concentration for 100% Bivalve Mortality	Mode of Action	Advantages	Disadvantages	Notes	Source
*Pseudomonas fluorescens* strain CL145A cells	Zequanox^®^	Zebra mussels		100	11 days	100 mg/L after 8 days *	Disrupt epithelial cells of digestive system lining, causing mortality	Registered EPA Office of Pesticides Programs Biopesticides and Pollution Prevention Division	Targeted for Zeba and Quagga mussels		[17,31,32,37,49]
								Highly selective to target species	Newer application method for open water systems		
									Cost prohibitive at scale		
Copper Sulfate Pentahydrate (cupric ion [Cu^++^])	EarthTecQZ^®^	Zebra mussels & Quagga mussels	0.18	0.18–1.0	96 h, repeated every 4–14 days	0.5–1.0 mg/L after 7 days *	Binds to gill membranes and causes tissue damage that interferes with gas exchange	Registered and approved by EPA as molluscicide (USEPA registration number 56576-1)	Can be toxic to non-target species in water and accumulated in soil		[31,32,50,51,52]
								Affects all life stages of mollusc	Water quality may diminish efficacy of treatment		
								Currently used as anti-fouling coating in industrial settings			
								Beginning field tests on other bivalves			
Chelated Copper- copper ions bound to amino acids	Cutrine^®^ Ultra	Zebra mussels	0.012	0.012–1.757	Maximum 96 h	NA	Binds to gill membranes and causes tissue damage that interferes with gas exchange	Registered and approved by EPA as algaecide, herbicide, and cyanobactericide (USEPA registration number 8959-53)	Not registered or approved by EPA for use as a molluscicide	72-h old trochophores achieved LC_50_ at value as low as 0.012 mg/L and LC_99_ at 0.047 mg/L within 2 h of exposure	[38,52,53]
										Adults achieved 96-h LC_50_ at 0.0352 mg/L and 96-h LC_99_ at 1.757 mg/L	
Copper (Cu) Ethanolamine	Natrix^®^	Quagga mussels	0.1	0.2–1.0	Maximum 96 h	NA	Binds to gill membranes and causes tissue damage that interferes with gas exchange	Registered and approved by EPA as molluscicide (USEPA registration number 67690-81)	Not-selective and negatively impacts non-target species	100% mortality never confirmed but 1 mg/L Natrix induced mortality of adult mussel within 48 h	[52,54,55]
								Quickly degraded and non-persistent in the environment			
								Only effects organisms with direct contact with application			
Potassium Chloride, KCl (K^+^)	Potash	Zebra mussels & Quagga mussels	NA	100–200	10 days	100 mg/L after 7–9 days *	Induces paralysis and alters behavior of closure behavior	Registered through site-specific regulatory exemption processes (Section 24© Special Local Needs exemption or Section 18 Emergency Exemption)	Not registered or approved by EPA for use as a molluscicide	Concentration of 50 mg/L achieved 100% mortality in same time frame with warmer water temperatures	[31,38,52]
							Promotes ciliostasis to reduce bivalve filtration rate	Non-toxic to species other than gill-breathing mollscs	Toxic to shelled organisms, such as native molluscs, crayfish, and zooplankton		
							Damages the gill epithelium to cause asphyxiation				
Microencapsulated PolyDADMAC	BioBullets^®^ SB 1000	Golden mussels & Wedge clams	12.5	400	48 h	500 mg/L after 2 days	Binds to gill membranes and interferes with gas exchange	Encapsulation bipasses bivalve closing respone	Newer technology still in toxicity testing		[4,37,41,56,57]
								Requires smaller concentration of active ingredient for similar efficacy	Unknown effects on non-target species		
								Highly selective to target species based on particle size and shape	Biocide shows poor environmental selectivity		
								New uses for existing active ingredients			
								Nearly 10 times more toxic than microencapsulated KCl			
Microencapsulated KCl	BioBullets^®^ SB 2000	Golden mussels & Wedge clams	125	3000	48 h	6000 mg/L after 2 days	Induces paralysis and alters behavior of closure behavior	Encapsulation bipasses bivalve closing respone	Newer technology still in toxicity testing		[37,56]
							Promotes ciliostasis to reduce bivalve filtration rate	Requires smaller concentration of active ingredient for similar efficacy	Unknown effects on non-target species		
							Damages the gill epithelium to cause asphyxiation	Highly selective to target species based on particle size and shape			
								New uses for existing active ingredients			
Esential Oils (EO)	EO, Eugenol (compound found in cinnamon, clove, and bay leaves)	Freshwater snail species: *Luminae acuminata*	2.55	2.55–10.73 Increased toxicity at elevated temperatures	24 h	NA	Reduces fecundity, hatchability, and survival of young	Minimal impact to non target species and surrounding environment	Insoluable or minimially soluable in water: require additional organinc solvent	2.55–10.73 mg/L for 24 h to achieve LC_50_	[4,35]
								Chemical compound complexity prevents resistance from developing	Easily oxidized in heat, light, and air		
									Cost prohibitive at scale		
									Newer technology still in toxicity testing		

## Data Availability

The original contributions presented in the study are included in the article/Appendix A; further inquiries can be directed to the corresponding author.

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
