# Peer review of "Chemical Treatments on Invasive Bivalve, Corbicula fluminea"

_animals, 2024, doi:10.3390/ani14121789_

Round 1

Reviewer 1 Report

Comments and Suggestions for Authors

The review entitled “Chemical Treatments on Invasive Bivalve, Corbicula fluminea” provides information about different chemical treatments used in open water systems to evaluate the efficacy of different molluscicides and their toxicity ranges and impact to water quality and non-target species. The review contains useful information and provides a good summery of the subject. It is well written and well structured. I recommend some minor revisions before acceptance by the journal

 Section 1 Introduction

1.3 Impact of Invasion

According to the review format, I would suggest to look for more recent bibliography and more according to some subjects like:

Line 92 and 93: In none of this citation (22, 23) it is talked about unionids. The authors can check actual bibliography in relation with the interaction between the Asian clam and unionids (Chiarello M., Bucholz, JR, McCauley, M., et al. 2022. Front. Microbiol, 13:800061.doi: 10.3389/fmicb.2022.800061; Modesto, V., Castro, P., Lopes-Lima, M., Antunes, C., Ilarri, M., Sousa, R. (2019). Science of the Total Environment 673 (2019) 157–164)

1.6 Treatment Selection

Line 176. Please remove in

Table 3.  Line 271. Please delete one dot of the table caption

Table 4. There is a mistake in the Chemical macroencapsulated KCl, in the last advantages part.

Section References:

The authors should write the bibliography in a unique style, e.g., the year of the publication, sometimes it is in bold and sometimes it is not.

Author Response

We are grateful for your expertise in reviewing this manuscript. We have revised according to your recommendations. Please see our responses below and the corresponding revisions/corrections highlighted/in track changes in the re-submitted files.

Reviewer 1 Comments

Authors’ Responses

1.3 Impact of Invasion

According to the review format, I would suggest to look for more recent bibliography and more according to some subjects like:

Line 92 and 93: In none of this citation (22, 23) it is talked about unionids. The authors can check actual bibliography in relation with the interaction between the Asian clam and unionids (Chiarello M., Bucholz, JR, McCauley, M., et al. 2022. Front. Microbiol, 13:800061.doi: 10.3389/fmicb.2022.800061; Modesto, V., Castro, P., Lopes-Lima, M., Antunes, C., Ilarri, M., Sousa, R. (2019). Science of the Total Environment 673 (2019) 157–164)

Yes, these two references are added to the text and literature accordingly. More elaborations on the interaction between Asian clam and unionids are added here as well. Reference numbering was adjusted throughout the manuscript.

1.6 Treatment Selection

Line 176. Please remove in

Revised accordingly.

Table 3.  Line 271. Please delete one dot of the table caption

Revised accordingly.

Table 4. There is a mistake in the Chemical macroencapsulated KCl, in the last advantages part.

After confirming with the source Calazans et al. 2013, the statement has been confirmed to be in reference to microencapsulated polyDADMAC: it is nearly 10 times more toxic than microencapsulated KCl. This advantage has thus been moved into the advantages for microencapsulated polyDADMAC.

Section References:

The authors should write the bibliography in a unique style, e.g., the year of the publication, sometimes it is in bold and sometimes it is not.

Revised accordingly.

Reviewer 2 Report

Comments and Suggestions for Authors

Corbicula

In general terms, I have the feeling that the authors did a good job of summarizing available literature on the control of Cf worldwide, but there are some statements and passages that could benefit from further revision. In particular, the manuscript at times gives the impression that eradication of the clam from its invasive range is feasible. In my opinion, this is a myth. Eliminating a small freshwater clam spread throughout millions of square kilometers across several continents is totally unrealistic. I suggest making it clear from the start that control strategies may be useful to prevent industrial installation pipe/filter clogging, or ecological impacts (e.g., protecting endangered native clams from excessive competition in isolated waterbodies), but complete eradication is highly unlikely.

The Asian clam Corbicula fluminea is a native aquatic species in Eastern Asia and Africa but has become one of the most ecologically and economically harmful invasive species in aquatic ecosystems in Europe, North America, South America...

I would suggest toning this statement down. There indeed have been a few reports of clogging of industrial/energy plants by the mussel in N America (and probably elsewhere as well), but these events are very rare, and none has been reported for South America. Its impacts of the local flora and fauna are most probably mixed, including both positive and negative effects.

The mode of life (buried in the sediment) of Cf should be mentioned, especially considering that the other major mollusc invasives (Dreissena, Limnoperna) live attached to hard substrata.

1.2. Asian Clams Dispersal in Massachusetts, USA  Previous studies reported that the Asian...

A detailed account of the dispersal of Cf in Massachusetts does not seem to be necessary for an overview of its control options worldwide

the costs to control Asian clam populations across the United States are estimated to be over $1 billion annually since 1980 [25]....

I suggest avoiding the inclusion of such unreliable estimates. Most of the data by Pimentel et al., although very highly cited, have been repeatedly shown to be extremely unreliable and biased.

Revise Table 1. Some entries need adjustment. For example, for Water Level Control the Pros column states that it does not require chemicals. While this is true, so do many of the other strategies listed, where this advantage is not mentioned. In short, whatever advantages are mentioned should be specific to the line in question, or mentioned throughout where applicable.

The fact that 90% of the work is based on USA cases and legislation should be clearly indicated, probably in the title.

Figure 2 is not necessary.

There seems to be some confusion between closed and open water systems. The qualification refers to the fate of the water used (often for cooling purposes): if it is reused (after filtering, condensation, etc.), then it is a closed system. If used water is discarded back into the waterbody and replaced by new raw water - then it is an open system. Please check throughout.

Table 3.

Mortality values reported for Cf exposed should be included. Post-exposure times (when and if analyzed) should also be mentioned (exposed clams can survive at the end of exposure but die some hours or days after the toxicant supply is discontinued).

Please specify whether that these data come from your own original lab study.

Table 4

Same comments as for Table 3

Author Response

We are grateful for your expertise in reviewing this manuscript. We have revised according to your recommendations. Please see our responses below and the corresponding revisions/corrections highlighted/in track changes in the re-submitted files.

Reviewer 2 Comments

Authors’ Responses

The Asian clam Corbicula fluminea is a native aquatic species in Eastern Asia and Africa but has become one of the most ecologically and economically harmful invasive species in aquatic ecosystems in Europe, North America, South America...

I would suggest toning this statement down. There indeed have been a few reports of clogging of industrial/energy plants by the mussel in N America (and probably elsewhere as well), but these events are very rare, and none has been reported for South America. Its impacts of the local flora and fauna are most probably mixed, including both positive and negative effects.

It is revised by delete “most”: The Asian clam Corbicula fluminea is a native aquatic species in Eastern Asia and Africa but has become one of the ecologically and economically harmful invasive species in aquatic ecosystems in Europe, North America, South America

The mode of life (buried in the sediment) of Cf should be mentioned, especially considering that the other major mollusc invasives (Dreissena, Limnoperna) live attached to hard substrata.

Added to section 1.3

 1.2. Asian Clams Dispersal in Massachusetts, USA  Previous studies reported that the Asian...

A detailed account of the dispersal of Cf in Massachusetts does not seem to be necessary for an overview of its control options worldwide

Cf in Massachusetts provides an example of invasiveness of this species, especially given the fact that climate change is facilitating its further spread. So, it is better to keep it here.

the costs to control Asian clam populations across the United States are estimated to be over $1 billion annually since 1980 [25]....

I suggest avoiding the inclusion of such unreliable estimates. Most of the data by Pimentel et al., although very highly cited, have been repeatedly shown to be extremely unreliable and biased.

It is revised as: Although the real cost is not known, the costs to control Asian clam populations across the United States are estimated to be over $1 billion annually since 1980 …

Revise Table 1. Some entries need adjustment. For example, for Water Level Control the Pros column states that it does not require chemicals. While this is true, so do many of the other strategies listed, where this advantage is not mentioned. In short, whatever advantages are mentioned should be specific to the line in question, or mentioned throughout where applicable.

The fact that 90% of the work is based on USA cases and legislation should be clearly indicated, probably in the title.

Revised accordingly

Figure 2 is not necessary.

According to Editor’s suggestion, Figure 2 needs to be kept here to meet the format requirement. 

There seems to be some confusion between closed and open water systems. The qualification refers to the fate of the water used (often for cooling purposes): if it is reused (after filtering, condensation, etc.), then it is a closed system. If used water is discarded back into the waterbody and replaced by new raw water - then it is an open system. Please check throughout.

Checked accordingly.

Further clarification was added to section 1.4 and the word “natural” was added throughout document to further specify that we are referring to natural, open water systems.

Table 3.

Mortality values reported for Cf exposed should be included. Post-exposure times (when and if analyzed) should also be mentioned (exposed clams can survive at the end of exposure but die some hours or days after the toxicant supply is discontinued).

Please specify whether that these data come from your own original lab study.

Revised accordingly. See details in the revised manuscript.

Table 4

Same comments as for Table 3

Revised accordingly. See details in the revised manuscript.